# Bond Behaviors of Steel Fiber in Mortar Affected by Inclination Angle and Fiber Spacing

**DOI:** 10.3390/ma15176024

**Published:** 2022-08-31

**Authors:** Xinxin Ding, Mingshuang Zhao, Hang Li, Yuying Zhang, Yuanyuan Liu, Shunbo Zhao

**Affiliations:** 1International Joint Research Lab for Eco-Building Materials and Engineering of Henan, North China University of Water Resources and Electric Power, Zhengzhou 450045, China; 2Collaborative Innovation Center for Efficient Utilization of Water Resources, North China University of Water Resources and Electric Power, Zhengzhou 450045, China

**Keywords:** hook-end steel fiber, bond performance, pullout test, inclination angle, fiber spacing, pullout load-slip curve

## Abstract

Considering the random orientation and distribution of steel fibers in concrete, the synergistic reinforcement of steel fibers on concrete is much complex than the bond of single fiber. It is meaningful to study the bond behavior of steel fiber during many actions. With the inclination angle of steel fiber to pullout direction and the fiber spacing as main factors, this paper carried out fifteen groups of pullout tests for hook-end steel fiber embedded in manufactured sand mortar. The inclination angle ranged from 0 to 60°, and the fiber spacing ranged from 3.5 mm to 21.2 mm. The characteristic pullout load-slip (*PL-S*) curve of steel fibers are given out after treating the original complete curves of each group test. The values of key points featured the debonding, peak and residual pullout loads and slips are determined from the characteristic *PL-S* curves. Based on a multi-index synthetical evaluation method, the nominal debonding strength, bond strength, residual bond strength and the debonding work, slipping work, and pullout work, as well as the debonding energy ratio, slipping energy ratio, and pullout energy ratio are analyzed. Results indicate that the bond performance represented by above indexes changes with the inclination angle and spacing of steel fibers. Except for the bond mechanism performing the same as aligned steel fibers by pullout test, the bond is dominated by the resistance of mortar to peeling off near pullout surface and scraping along pullout direction. When the inclination angle is over 15° or 30°, the bond performance is generally decreased, due to the peeling off of mortar on surface of transversal section with a certain depth. When the fiber spacing is over than 5 mm, the bond performance becomes worst due to the scraping out of mortar along with the slip of steel fibers.

## 1. Introduction

With short and random distributed steel fibers, a composite material is produced to be the steel fiber reinforced concrete (SFRC) [1,2,3]. This promotes the post-cracking performances of conventional concrete [4,5,6,7]. From the viewpoint of meso-mechanism, the bond of steel fiber is essential to the reinforcement effect, and the steel fibers bridging cracks of concrete matrix enhance the post-cracking performance of SFRC [2,4,7]. In other words, the reinforcement effect depends on the bond which may be affected by the orientation and distribution of steel fibers in concrete [8,9,10].

The bond mechanisms have attracted considerable attention in active research [11,12]. Many studies based on the pullout test of single or multi aligned fibers have been performed in order to understand the bond behaviors of steel fibers in concrete or mortar. Except for the geometry of steel fiber, rough surface of steel fiber [13] and high-performance cementitious matrix [10,14,15] benefit to the bond performance. By purposefully reinforcing the strength or the toughness of cementitious matrix, or both simultaneously, different deformed steel fibers can be selected with different bond behaviors [16,17,18].

Normally, the inclination of steel fibers avoids the direct pullout of steel fibers from cementitious matrix. These benefits to the steel fibers worked together with the cementitious matrix [13,19,20,21]; however, it leads a potential rupture of fibers and matrix with higher tensile stresses [22,23,24,25,26,27,28]. In this aspect, the influence of the inclination angle on the pullout behavior of steel fibers depends on the types, size and embedded length of steel fiber and the matrix strength [24,29]. As report by Chun [13], a higher bond strength with a lower energy absorption capacity was observed for the straight steel fibers with inclination angle to 45° in the ultra-high-performance concrete. The study of Huang [19] obtained that the ultimate pullout load and pullout energy increased with the increase in inclination angle from 0° to 45° for the brass-coated straight steel fiber embedded in reactive powder concrete. Similarly, other investigations also indicated that the pullout load reached the peak for straight steel fibers with inclination angle of 30° [20,21], 30° or 45° [22] and 45° [30], regardless of the fiber size and the matrix strength, although the consumption energy during pullout could keep increasing as angles to 60° [20].

Compared to that of the straight steel fibers, the pullout behavior of hook-end fibers presents a less sensitivity to the inclination angle. With the same length of 13 mm and diameter of 0.2 mm, when the inclination angle increased from 0° to 30° and 45°, the bond strengths of straight fiber increased by 19.2% and 52.9%, while those of hook-end fiber increased by 10.3% and 16.2%. Meanwhile, the bond strengths of hook-end fiber with a length of 25 mm and a diameter of 0.35 mm increased by 13.6% and 26.1%, respectively [27]. For both the hook-end steel fiber with a length of 60 mm and diameter of 0.75 mm and the straight fiber cut from the hook-end fiber, when the inclination angle increased from 0° to 30°, the peak pullout load of straight fiber with the embedded lengths of 20 mm and 30 mm increased by 124% and 31.2%, while those of hook-end fiber increased by 15% and 7%, the peak slip increased with the inclination angle [9]. The peak pullout load of hook-end steel fiber with a length of 30 mm and a diameter of 0.38 mm was similar with the inclination angels ranged from 0° to 30°, whereas it increased with the angle and reached the maximum around the inclination angle of 20° for hook-end steel fiber with a length of 60 mm and a diameter of 0.9 mm [24]. Wang [25] has carried out the pullout tests of hook-end steel fiber with a length of 35 mm and a diameter of 0.55 mm embedded in concrete with water to cement ratio of 0.49 at the inclination angles of 0°, 30°, 45° and 60°. The result showed that the bond strength decreased by 27.3% while the peak slip gradually increased with the inclination angle increased from 0° to 60°.

At present, few studies concerned the fiber group effect by pullout test using multi fibers. Feng [31] held that the group effect of the hook-end steel fiber weakened the bond between the fibers and magnesium phosphate cementitious matrix when the spacing of the fibers changed from 16 mm to 6 mm. Kim and Yoo [32,33] reported that approximately 30% lower bond strengths were obtained from the specimens with multiple fibers compared to those with a single straight fiber, a hook-end fiber or a twisted fiber. The average bond strengths of the hook-end and twisted steel fibers were improved by decreasing the fiber spacing up to 1 mm, corresponding to a volume fraction of 7% based on an assumption of perfect fiber distribution. Thus, the synergistic reinforcement mechanism of steel fibers on concrete needs to be enriched.

The pullout performances of different deformed steel fibers with hook-end, crimped, indentation, milled and large-end in the mortars with different strengths and aggregates have been systematically studied in previous works [14,16]. A multi-index synthetical evaluation method was built based on the key points of the characteristic pullout load-slip (*PL-S*) curve to quantitatively evaluate the bond strengths, energy dissipation abilities and toughness of steel fiber, which corresponded to the loading cases of cracking resistance, normal serviceability and ultimate bearing capacity of SFRC, respectively. Results showed that the whole bond performance after the slipping of steel fiber attributed from the increasing strength of mortar with manufactured sand, although distinct reinforcing and toughening effects presented on the bond of different fibers. Therefore, the synergistic working of steel fibers in the concrete matrix should be realized to only or simultaneously improve the strength and toughness of SFRC.

Based on above discussion of previous studies, three series of pullout tests for the hook-end steel fiber embedded in manufactured sand mortar were carried out in this study. The pullout behavior of steel fibers in groups with the implications of the inclination angle, the angle hybrid and the fiber spacing were examined. Two series were tested for steel fibers with the inclination angles and the angles hybrid of 0°, 15°, 30°, 45° and 60°. One series were tested with the number of aligned fibers of 1, 2, 9, 16 and 25. Subsequently, several important pullout parameters [14,16], including the debonding strength, bond strength, residual bond strength and the debonding work, slipping work and pullout work, as well as the debonding energy ratio, slipping energy ratio and pullout energy ratio, were analyzed.

## 2. Experimental

### 2.1. Preparation of Mortar

The raw materials used for mortar included ordinary silicate cement, fly ash, manufactured sand, high-performance polycarboxylate water reducer and tap water. The manufactured sand was made of limestone with a fineness modulus of 2.73 and a stone powder content of 7.3%. Properties of the raw materials are the same as previous reports [14,16]. The self-compacting workability of the fresh mortar was contributed from the addition of fly ash as mineral admixture, to ensure the accuracy of the fibers position in the mortar specimen. Details of mixture proportion and workability of mortar is summarized in Table 1.

A planetary-type mortar mixer was used for the mixing of fresh mortar. The water and water reducer, the cement and fly ash were added in the mixing pot to mix for 30 s, then the manufactured sand was evenly added to continuously mix for 30 s. After that, the fresh mortar was mixed for another 30 s, and cast into the molds of specimens.

### 2.2. Details of Specimen

The steel fiber was hook-end shaped with a length of 29.8 mm, a diameter of 0.5 mm and a tensile strength *f*_sf_ of 1150 MPa.

A total of sixty dog-bone shape specimens were used in the study. Three series with fifteen group of pullout test were designed; each group had four specimens—refer to the pullout specimen design in the specification of China code CECS 13 [34]. Figure 1a presents the geometric details of specimen for Series IA with two pairs of inclined steel fibers. The specimens, marked as IA0, IA1, IA2, IA3 and IA4, respectively corresponded to the inclination angle of 0°, 15°, 30°, 45° and 60°. Figure 1b presents the geometric details of specimen for Series HIA with two pairs of steel fibers. One pair of steel fiber was arranged with inclination angle of 0°, 15°, 30°, 45° and 60°, respectively, while other pair aligned horizontally along the pullout direction. The specimens were respectively marked as HIA0, HIA1, HIA2, HIA3 and HIA4. Figure 1c presents the geometric details of specimen for Series NA with different number of aligned steel fibers. The area of 16 mm × 16 mm in the center of the cross section is designed as the area for fiber placement. The fiber number are designed as 1, 2, 9, 16 and 25, corresponding to the fiber center-to-center spacing *L*_sf_ of infinite, 21.2 mm, 7.5 mm, 5 mm and 3.5 mm. The specimens are marked as NA0, NA1, NA2, NA3 and NA4, successively. Meanwhile, the specimens of Series IA and Series HIA with inclination angle of 0° were used as the ones with 4 aligned steel fibers in spacing of 15 mm.

The pull-out test of steel fibers in mortar was realized by different bond length embedded in the two parts of specimen. One part was used to fix steel fiber with longer length in the mortar, while the steel fibers with shorter length in other part would be pulled out of the mortar. The embedded length and fixed length of steel fibers for Series IA, HIA and NA are given in Table 2.

### 2.3. Test Method

The specimen was formed in the dog-bone shape mold [14,16]. In order to minimize the disturbance of demolding on the bond between the fibers and the mortar matrix, the specimen was demolded after cast for 2 days, and placed into the standard curing box for next 26 days before testing. The standard curing box has a temperature of 20 ± 2 °C and humidity larger than 95%.

As in previous studies [14,16], the specimen was stretched using an electronic universal testing machine under a loading speed of 0.3 mm/min to obtain the completely tested pull-out load vs. slip curve (*PL-S* curve). Then, as shown in Figure 2, the four tested *PL-S* curves of each group specimens were processed to be a characteristic *PL-S* curve. The detailed processing method for the ascending portion and descending portion of the four tested *PL-S* curves are presented in previous study [16]. The feature values were extracted at the key points, which represent the slope changing in ascending portion, at peak and in descending portion of the characteristic *PL-S* curve.

Accompanied with each series of specimens, three mortar prisms with dimension of 40 mm × 40 mm × 160 mm were poured for determining its flexural and compressive strengths as per China code GB/T17671 [35]. Due to the difference of casting time and environment, a differential of strength exists in different series of specimens, even though they all have the same composition and test age. The compressive and flexural strengths of mortar used for Series IA, HIA and NA were 75.8 MPa and 11.13 MPa, 69.9 MPa and 8.35 MPa, 77.8 MPa and 12.38 MPa with standard variations of 6.6 MPa and 1.11 MPa, 5.8 MPa and 1.24 MPa, 3.89 MPa and 1.24 MPa, respectively.

## 3. Test Results and Analyses

### 3.1. Failure Modes

Two failure modes featured by the fiber pullout and the mortar spalling were observed in this study.

For series IA, the steel fibers of IA0 and IA1 were pullout with straightened hook-end; the steel fibers of IA2, IA3 and IA4 were pullout with straightened hook-end accompanied by the mortar spalling surrounded the fibers. As presented in Figure 3, the volume of mortar spalling significantly increases with the inclination angles. Although a similar peeling off area of the mortar presented on specimens with steel fibers inclined at angle 45° and 60°, a larger peeling off depth happened on the specimens with steel fibers at greater inclination angle. This phenomenon is also reported in the reference [23]. With the increase in inclination angle of steel fiber, the peeling off force perpendicular to transversal section increased during the pull-out of steel fibers, which would be much more increased with the process of steel fibers straightened. This results in the tensile stress that could be over the tensile strength of mortar. If that occurs, the mortar near surface of transversal section will peeling off.

Figure 4 presents the typical failure mode of Series HIA. The spalling areas of mortar near the straight and inclined fibers are highlighted as red and blue circles, respectively. All the steel fibers of HIA0, HIA1 and HIA2 were pullout with straightened hook-end. The aligned fibers of HIA3 and HIA4 were pullout with the straightened hook-end, while the inclined steel fibers were pullout with straightened hook-end accompanied by the mortar spalling surrounded the fibers. This also indicted the effect of inclination angle on the bond performance of steel fibers in mortar. With the increase in inclination angle, the straightening degree of the hook-end decreases, while the spalling volume of mortar significantly increases.

Figure 5 presents the typical failure mode of Series NA. All steel fibers of NA0, NA1, NA2 and NA3 were pullout with straightened hook-end, while the mortars of NA4 peeled off accompanied with slightly straightened hook-end of steel fibers. The tensile strength of the mortar near the fiber end is not enough to resist the stress transmitted by the fiber to the matrix. This indicates a rational spacing among steel fibers is necessary to ensure a sufficient surrounding mortar which can provides anchorage shear resistance of the interface between steel fiber and mortar.

Based on the above description, in condition of the mortar with a compressive strength around 74.5 MPa, the steel fibers will be pullout with the straightened hook-end. If the inclination angle of steel fiber is over than 30°, the mortar will be peeled off from the transversal section. When the fiber spacing is 3.5 mm, the mortar surrounded the steel fibers can be scraped out during the pulling out of steel fibers. This indicates the bond performance will be affected by the orientation and distribution of steel fiber in concrete matrix.

### 3.2. The Characteristic Pullout Load-Slip Curve

Figure 6 presents the characteristic pullout load-slip (*PL-S*) curves of Series IA, HIA and NA. As explained in previous studies [14,16], the bond performance at debonding, peak and residual cases can be expressed by the corresponding point at the characteristic *PL-S* curve with a change of slope in ascending portion, at peak and in descending portion. They respectively represent the debonding load *P*_d_ and slip *s*_d_, the peak load *P*_p_ and slip *s*_p_, and the residual load *P*_r_ and slip *s*_r_. Values of them are listed in Table 3.

In Series IA, the curve of IA1 with the inclination angle of 15° almost coincides with that of IA0. With the inclination angle increased from 15° to 60°, the slope of the ascending portion decreases, the *P*_p_ decreased about 45.2%, while the peak-slip *s*_p_ increased about 26.2%. The regularity is also reported by the reference [25]. The *P*_d_ increased 13.4% with the inclination angle increased to 15°, then obviously decreased 78.8% with the angle continuously increased to 60°. The *P*_r_ increased 29.5% with the inclination angle increased to 30°, then decreased 20.3% with the angle continuously increased to 60°. Both slips *s*_d_ and *s*_r_ presented the decrease trends with the increased inclination angle. The fluctuation of the descending portion of the curve gradually increases, and the area under curve gradually decreases. The variation of peak-slip is consistent with the reference [36]. Similar regularities are observed in the characteristic curves of Series HIA, the change degree of which almost reduced by half, due to a pair of steel fibers aligned to the pullout direction of series HIA.

In Series NA, the complete *PL-S* curves of NA0, NA1, NA2 and NA3 are obtained with failure mode of fibers pullout. The *P*_d_, *P*_p_ and *P*_r_ of NA0, NA1 and NA2 are positively correlated with the number of fibers of 1, 2 and 9, successively. The similar slips *s*_d_, *s*_p_ and *s*_r_ of NA0, NA1, NA2 were observed, while, the *P*_d_ and *P*_p_ of NA3 are significantly higher than the load values of NA0 multiplying the fiber number. NA4 only got the ascending portion of the curves, due to the failure mode of mortar peeled off.

### 3.3. Bond Strengths and Strength Ratio

The nominal debonding strength *τ*_d_, bond strength *τ*_max_ and residual bond strength *τ*_r__es_ are calculated using Formulas (1)–(3), and the test values of them are present in Figure 6. To reflect the debonding resistance of steel fiber from mortar and the loss rate of bond strength in the descending portion, the nominal fiber utilization efficiency *u*_sf_, the nominal strength ratios *u*_de_ and *u*_res_ are calculated using Formulas (4)–(6), and the test values of them are listed in the Table 4.
(1)$$\tau_{d}=\frac{P_{d}}{n\pi d_{f}\left(l_{f,\mathrm{em}}-s_{d}\right)}$$
(2)$$\tau_{\max}=\frac{P_{p}}{n\pi d_{f}\left(l_{f,\mathrm{em}}-s_{p}\right)}$$
(3)$$\tau_{\mathrm{res}}=\frac{P_{r}}{n\pi d_{f}\left(l_{f,\mathrm{em}}-s_{r}\right)}$$
(4)$$u_{\mathrm{sf}}=\frac{4P_{p}}{n\pi d_{f}f_{\mathrm{sf}}}$$
(5)$$u_{\mathrm{de}}=\frac{\tau_{d}}{\tau_{\max}}$$
(6)$$u_{\mathrm{res}}=\frac{\tau_{\mathrm{res}}}{\tau_{\max}}$$

Figure 7a shows the variations of *τ*_d_, *τ*_max_ and *τ*_res_ of Series IA with the inclination angle. Slight inclination of steel fibers benefits to the resistance of debonding and the residual bond. This leads that *τ*_d_ and *τ*_res_ reach the maximum at the angle of 15° and the angle of 30° with an increment of 14% and 17%, respectively, compared to that of IA0. However, the *τ*_max_ trends to decrease with the increase in inclination angle from 0° to 15°, and decreases linearly with the inclination angle increased from 15° to 60°. The *τ*_max_ of IA4 is about 44% lower than that of IA0. This is similar to the pullout test result of single hook-end steel fiber [31].

With the increase in the inclination angle, the risk of mortar cracking increases and the fiber utilization rate decreases. The nominal fiber utilization efficiency *u*_sf_ decreases by 45.2% with the angle increased to 60°. The nominal strength ratio *u*_de_ increases by 15% with the inclination angle increased from 0° to 15°, and then decreases by 64% with the angle increased from 15° to 60°. This illustrates that the debonding resistance is sensitive to the inclination angle. The nominal strength ratio *u*_res_ increases by 38% with the inclination angle increased from 0° to 30°, and then changes a little with the continuous increase in the angle. This indicates that the loss rate of bond strength can be reduced with a larger inclination angle of steel fiber, due to the better bond retention by the compressive action of peeling off force perpendicularly on the steel fibers.

Figure 7b shows the variations of *τ*_d_, *τ*_max_ and *τ*_res_ of Series HIA with the hybrid inclination angle. Compared with Series IA, half of steel fibers aligned to the pullout direction. This lightened the effect of inclination angle on the bond of steel fibers. With the hybrid action of aligned and inclined steel fibers, the *τ*_d_ of HIA2 and HIA3 are 24% and 21% higher than that of HIA0. Although the *τ*_max_ still trends to decrease with the increase in the inclination angle, the decrement becomes slower. The decrement is 25% with the inclination angle increased from 15° to 60°, which is 56.8% that of the IA4. Therefore, the reduction of *τ*_max_ comes from the decreased bond strength of the inclined steel fiber in Series HIA, no hybrid effect exists among inclined and aligned steel fibers. A slight increase in the *τ*_res_ appears with the increase in inclination angle. This is due to the better bond retention of inclined steel fiber during the pullout.

Except for the large inclination angle of 60°, the smaller angle benefits to the debonding resistance and the retention of residual bond. The strength ratio *u*_de_ increases 50.7% with the inclination angle from 0° to 30°, and then decreases 35.7% with the inclination angle continuously increased from 30° to 60°. The strength ratio *u*_res_ increases 31.4% with the inclination angle from 0° to 60°. The nominal fiber utilization efficiency *u*_sf_ decreased by 27.3% with the angle increased to 60°. It is almost half decrease rate compared with the regularity of *u*_sf_ for series IA. It also illustrates that no hybrid effect exists among inclined and aligned steel fibers.

Figure 7c shows the variations of *τ*_d_, *τ*_max_ and *τ*_res_ of Series NA with the fiber number. The influence of fiber number on bond strength essentially relates to the influence of fiber spacing. There are slight decreases of the *τ*_d_, *τ*_max_, *τ*_res_, *u*_sf_, *u*_de_ and *u*_res_ of NA1 compared with those of NA0. This may be due to the eccentric pullout on the two steel fibers during the loading process. In addition, the reduction rate of the bond strengths of multiply fibers compared with single fiber is smaller than that with reported in the references [32,33]. This may attribute to the different cementitious matrix and pull-out test method. When the fiber number increased to 9 for NA2, the *τ*_d_, *τ*_max_, *u*_sf_ and *u*_de_ are basically equal to those of NA0, while the *τ*_res_ and *u*_res_ slightly decrease. When the fiber number reached to 16 for NA3, the *τ*_d_, *τ*_max_, *u*_sf_ and *u*_de_ increase by 20.2%, 8.1%, 8.1% and 11.7% compared with those of NA0, respectively. This indicates that a group effect of parallel fibers with fiber spacing no less than 5 mm benefits to the bond performances. Therefore, an interaction exists among steel fibers in concrete matrix if the steel fibers are uniformly distributed in parallel with a volume fraction over 0.78% (corresponding to *L*_sf_ of 5 mm).

The values of *τ*_d_, *τ*_max_ and *τ*_res_ of IA0 and HIA0 are also presented in the Figure 6c. Ignoring the effect of embedded length, comparing the values of *τ*_d_, *τ*_max_ and *τ*_res_ of IA0, HIA0, NA0, NA1 and NA2, all the bond strengths increase with the flexural strength of mortar. The result is consistent with the previous study [14].

In addition, to get the real bond performance of steel fiber without influenced by the eccentric loading, a reasonable number of fibers should be used in the pullout test [32,33,37,38]. Comprehensively considering the range and test accuracy of the test system in this study, four steel fibers symmetrically arranged to section centroid is a better of chose.

Considered the pullout process of inclined steel fibers, the bond of steel fiber to mortar not only comes from the shear stress on interface, but also from the perpendicular pressure on interface, as shown in Figure 8. The former is the same as that of aligned steel fiber, which comes from the chemical adhesion and the mechanical fraction on interface between fiber and mortar and the anchorage of hook-end [14,16]. The latter comes from the component force of pullout load, which directly relies on the pullout load and the peeling off resistance of mortar. If the component force of pullout load is over the peeling off resistance of mortar, the mortar will be peeled off. This leads to broken off mortar from the surface of transversal section to the inner along steel fiber, and results in a shortening of the bond length of steel fiber. The final presentation of the nominal bond strengths by Formulas (1) to (3) will be decreased. With the increase in inclination angle, the component force of pullout load increases to rise the possibility of the peeling off of mortar, as presented in Figure 3 and Figure 4.

### 3.4. Bond Works

The debonding work *W*_d_, the slipping work *W*_p_ and the pullout work *W*_r_ are used for evaluating the energy dispersion during the bond-slip process. They are the areas under the characteristic *PL-S* curve with the slip from origin to the slips at debonding, peak and residual points, respectively. Formulas are listed as follow:
(7)$$W_{d}=\int_{0}^{S_{d}}Pds$$
(8)$$W_{p}=\int_{0}^{S_{p}}Pds$$
(9)$$W_{r}=\int_{0}^{S_{r}}Pds$$

Table 5 presented the test values of *W*_d_, *W*_p_ and *W*_r_. In Series IA, the *W*_d_ increases by 30% with the inclination angle from 0 to 15°, and then decreases by 92% with the inclination angle continuously increased from 15° to 60°. The *W*_p_ has no change with the inclination angle from 0 to 15°, and then decreases with the increase in inclination angle. The *W*_p_ of IA4 is 34% lower than that of IA0. The *W*_r_ has a linear decrease with the increase in inclination angle. The *W*_r_ of IA4 is 64% lower than that of IA0. Therefore, when the inclination angle of steel fibers is larger than 15°, the bond energy will be sustainably decreased.

In Series HIA, the *W*_d_ obviously increases with the inclination angle of steel fibers, while the *W*_p_ and *W*_r_ trend to decrease with the increase in the inclination angle. This is consistent to the influence of the hybrid action of inclined to aligned steel fibers on the bond strength.

The influence of fiber spacing on the bond energy can be reflected by the bond works per single fiber. Therefore, the bond works *W*_d_, *W*_p_ and *W*_r_ of Series NA divided the fiber number *n* are listed in Table 5. With the decrease in fiber spacing from 22.2 mm to 5 mm, the *W*_d_/*n*, *W*_p_/*n* and *W*_r_/*n* increase 263%, 22% and 47%, respectively. This means an improving effect of reasonable fiber number on the bond energy. In addition, the lower bond works of NA1 indicate that the eccentric loading of pullout test for two steel fibers should be avoided to get real bond performance.

### 3.5. Energy Ratios

The debonding energy ratio *R*_d_, the slipping energy ratio *R*_dp_ and the pullout energy ratio *R*_pr_ of hook-end steel fiber are used to represent the energy dissipation ability in different stages of the characteristic *PL-S* curve from the origin to the debonding point, from the debonding point to the peak point and from the peak point to the residual point [14]. Formulas can be written as follow,
(10)$$R_{d}=\frac{W_{d}}{W_{r}}$$
(11)$$R_{\mathrm{dp}}=\frac{W_{p}-W_{d}}{W_{r}}$$
(12)$$R_{\mathrm{pr}}=\frac{W_{r}-W_{p}}{W_{r}}$$

As shown in Figure 9a, the *R*_d_ increases by 65% with the inclination angle from 0 to 15°, and then decreases by 84% with the inclination angle continuously increased to 60°. *R*_dp_ increases 110% and *R*_pr_ decreases 36% with the inclination angle increased from 0 to 60°. This indicates that the energy dissipation ability increases before peak-slip and decreases afterward. To be applied for the concrete structures, the inclined steel fiber is favorable to the crack control at the normal serviceability, while less toughness and lower energy dispersion ability at the ultimate bearing capacity.

As presented in Figure 9b, the *R*_d_ of Series HIA increases with the inclination angle of steel fiber. The *R*_d_ of HIA4 is 11.9 times that of HIA0. This indicates the hybrid effect of inclined and aligned steel fibers is favorable on the energy dissipation before cracking of concrete matrix. The *R*_dp_ and the *R*_pr_ of Series HIA present the similar changes to those of Series IA with the increase in inclination angle of steel fiber; however, the changes become slowly due to half number of steel fibers were inclined. The *R*_dp_ increases by 40% with the inclination angle from 0 to 45°, and then decreases by 16% afterward. The *R*_pr_ decreases by 23.2% with the inclination angle from 0 to 60°. This once again indicates that the energy dissipation ability increases before peak-slip and decreases afterward with the inclination angle of steel fibers.

As shown in Figure 9c, with the fiber spacing decreased from 22.5 mm to 5 mm, the *R*_d_ increases by 146%, the *R*_dp_ decreases by 37% and the *R*_pr_ increases by 13%. This indicated that the energy dissipation ability increases at the debonding process and the residual bond stage, which is consistent to the strengthening of tensile strength and toughness of steel fiber reinforced concrete. Meanwhile, it shows that steel fibers in concrete matrix should keep a reasonable spacing to develop their reinforcing contribution. Therefore, the content of steel fiber in concrete should be optimized. At the same time, the variation of energy ratios between NA0 and NA1 can be concluded to the validity of the pullout test with two steel fibers.

## 4. Conclusions

The effect of inclination angle of steel fiber, hybrid inclined and aligned steel fibers and fiber spacing on the bond behaviors of steel fibers in manufactured sand mortar were experimentally studied by the pullout test, results are discussed based on the bond mechanisms of hook-end steel fiber. Conclusions can be drawn as follows:

The nominal debonding strength, debonding work and debonding energy ratio reached the maximum when the inclination angle was 15° and decrease afterward, while the nominal bond strength and the slipping work had slight changes at the inclination angle of 15° and decrease afterward. The nominal residual bond strength reached its maximum at the inclination angle of 30° and decrease afterward. The slipping energy ratio increases, and the pullout work and the pullout energy ratio decrease with the increase in inclination angle. This indicates that the inclination angle of steel fibers has obvious effect on the reinforcing effect of steel fiber on concrete matrix.

Compared with the bond behaviors of specimens with couples of inclined steel fibers, no hybrid effect among the inclined and aligned fibers are observed on the nominal debonding strength and the nominal bond strength. However, a favorable hybrid effect presented on the nominal residual bond strength and the debonding energy ratio. This indicates the different bond performance can be provided by steel fiber in different orientation.

The nominal debonding strength, bond strength and residual strength increases with the decrease in fiber spacing no less than 5 mm. The scraping failure of surrounding mortar appeared in condition of much smaller spacing of steel fibers, due to insufficient mortar resists the scraping force during pullout. Moreover, specimen with fiber number from 4 to 9 in this study are suitable for study the bond mechanism between the fiber and mortar matrix. Comprehensively considering the range and test accuracy of the test system in this study, four steel fibers symmetrically arranged to section centroid is a better choice.

## Figures and Tables

**Figure 1 materials-15-06024-f001:**
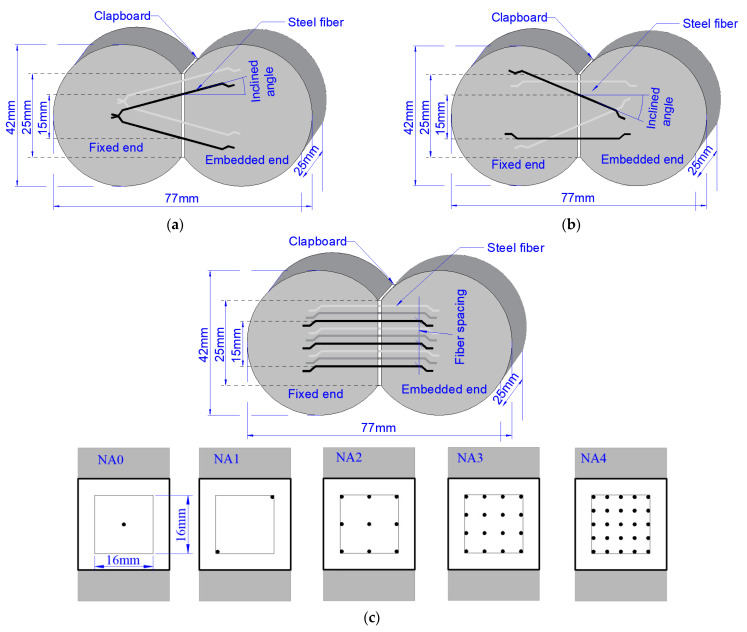
The geometric details of specimen for the pullout test. (**a**) Series IA (**b**) Series HIA (**c**) Series NA.

**Figure 2 materials-15-06024-f002:**
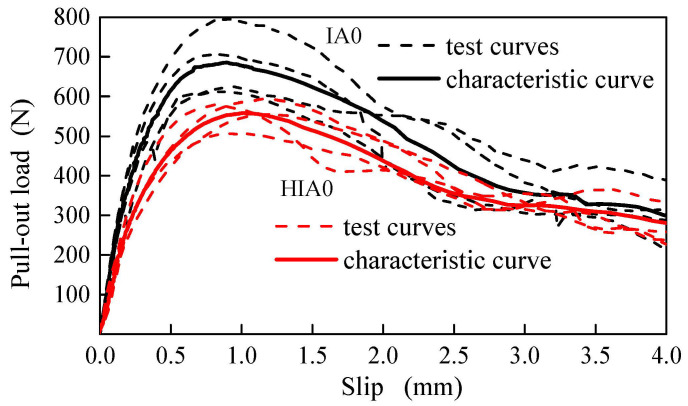
Two characteristic *PL-S* curves from the test curves of two groups of specimens.

**Figure 3 materials-15-06024-f003:**
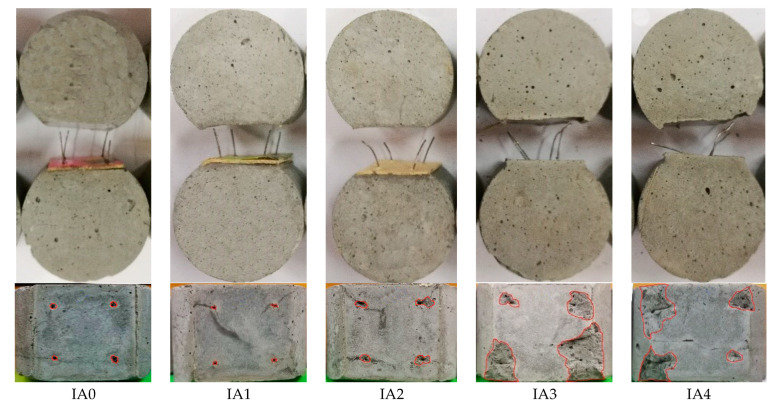
Typical failure mode of Series IA.

**Figure 4 materials-15-06024-f004:**
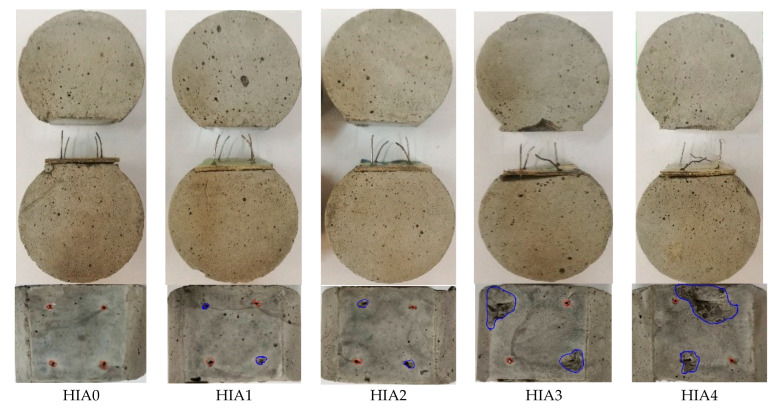
Typical failure mode of Series HIA.

**Figure 5 materials-15-06024-f005:**
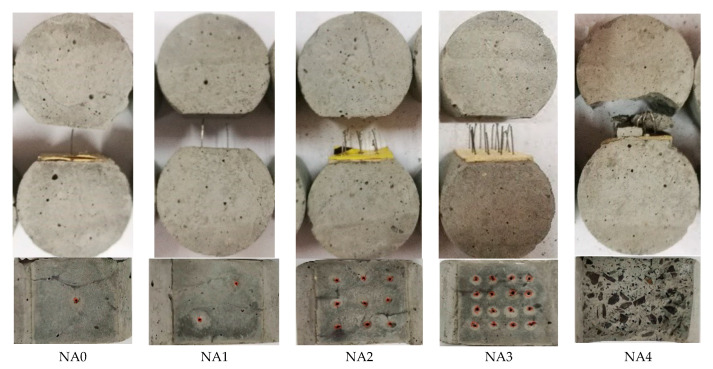
Typical failure mode of Series NA.

**Figure 6 materials-15-06024-f006:**
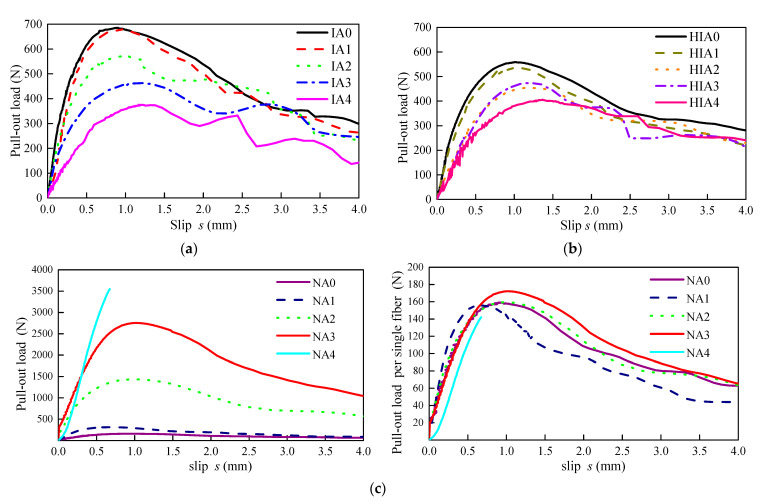
The characteristic PL-S curves. (**a**) Series IA, (**b**) Series HIA and (**c**) Series NA.

**Figure 7 materials-15-06024-f007:**
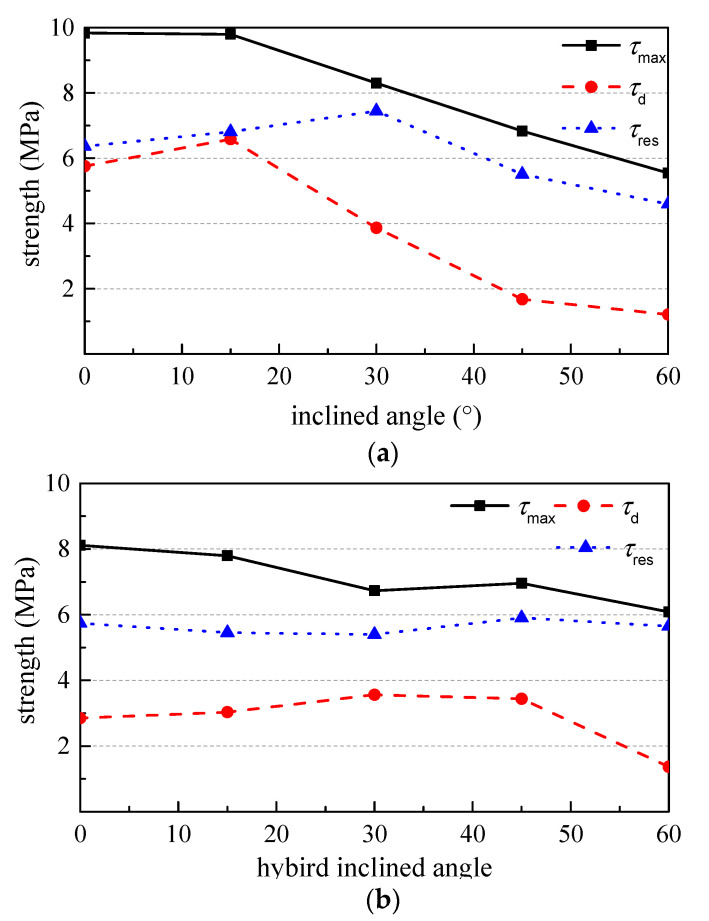
The bond strengths. (**a**) Series IA, (**b**) Series HIA and (**c**) Series NA.

**Figure 8 materials-15-06024-f008:**
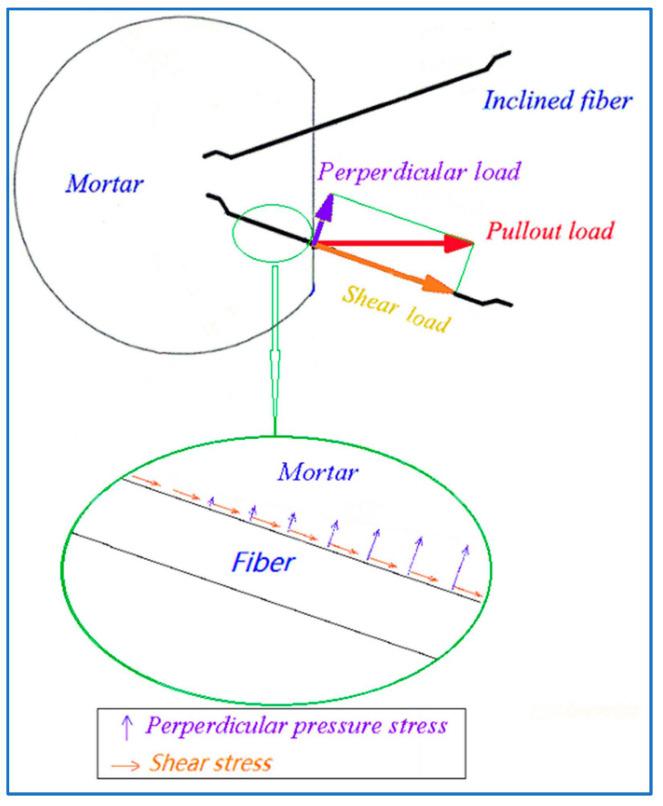
Actions on bond interface of inclined steel fiber.

**Figure 9 materials-15-06024-f009:**
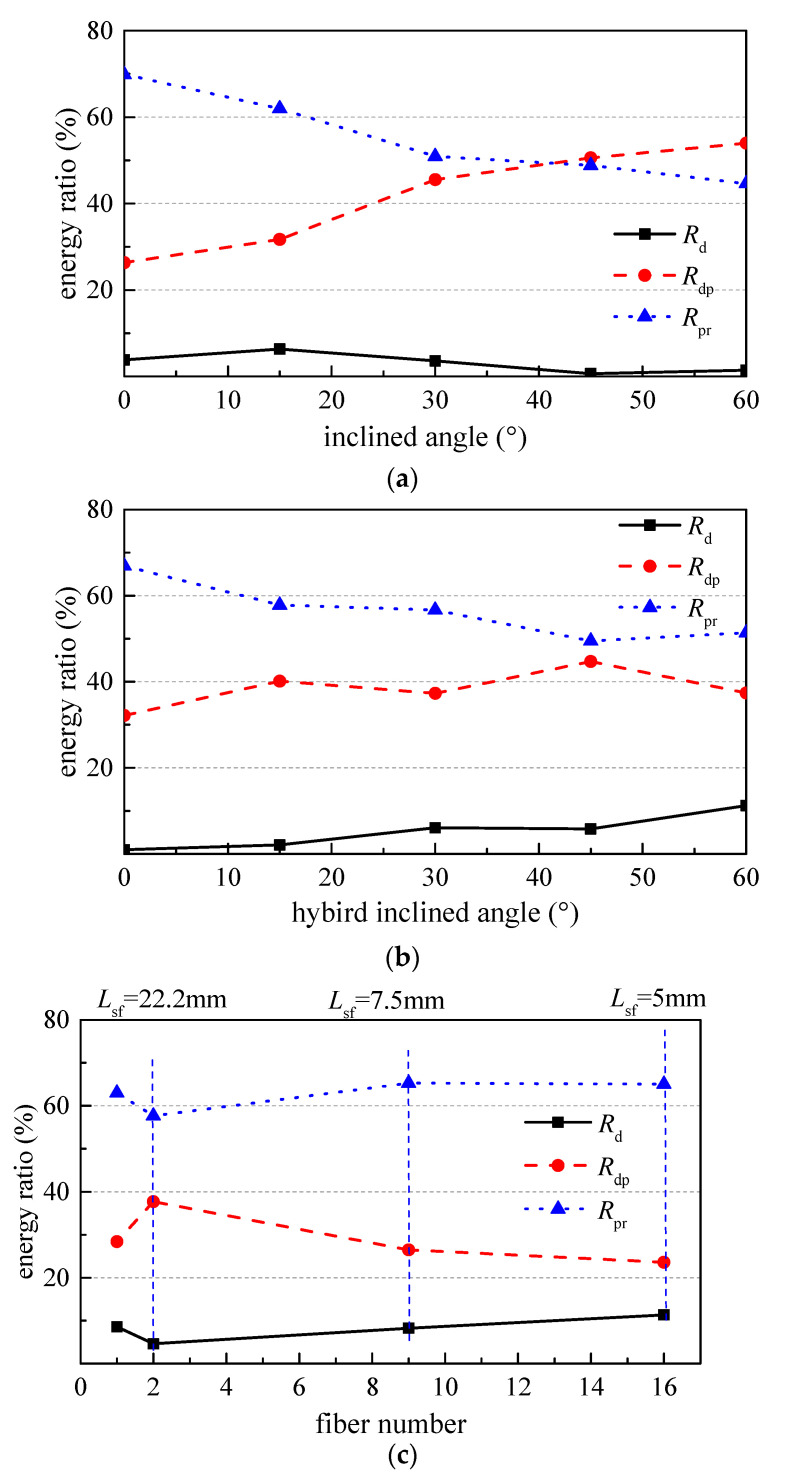
Influence of various factors on the bond energy ratios. (**a**) Series IA, (**b**) Series HIA and (**c**) Series NA.

**Table 1 materials-15-06024-t001:** Mix proportion and workability of mortar.

Mix Proportion	Water to binder ratio *w/b*	0.31
	Water (kg/m^3^)	277.9
	Cement (kg/m^3^)	627.6
	Fly ash (kg/m^3^)	269.0
	Manufactured sand (kg/m^3^)	1110.4
	Water reducer (kg/m^3^)	8.06
Micro slump flow (mm)		250

**Table 2 materials-15-06024-t002:** Details of the specimen designed for the pullout test.

Trials	Embedded Length (mm)	Fixed Length (mm)	Fiber Number	Inclination Angle (°)				Fiber Spacing (mm)	Influence Factor
				Fiber 1	Fiber 2	Fiber 3	Fiber 4		
IA0	10	18.8	4	0	0	0	0	15	Inclination angle for two pairs of steel fibers
IA1	10	18.8	4	15	15	15	15	/	
IA2	10	18.8	4	30	30	30	30	/	
IA3	10	18.8	4	45	45	45	45	/	
IA4	10	18.8	4	60	60	60	60	/	
HIA0	12	16.8	4	0	0	0	0	15	Inclination angle for a pair of steel fibers
HIA1	12	16.8	4	0	15	0	15	/	
HIA2	12	16.8	4	0	30	0	30	/	
HIA3	12	16.8	4	0	45	0	45	/	
HIA4	12	16.8	4	0	60	0	60	/	
NA0	11	17.8	1	0	0	0	0	/	Fiber spacing
NA1	11	17.8	2					21.2	
NA2	11	17.8	9					7.5	
NA3	11	17.8	16					5	
NA4	11	17.8	25					3.5	

**Table 3 materials-15-06024-t003:** Test values at key points of the characteristic *PL-S* curves.

Trials	Peak Point		Debonding Point		Residual Point	
	*P*_p_ (*N*)	*s*_p_ (mm)	*P*_d_ (*N*)	*s*_d_ (mm)	*P*_r_ (*N*)	*s*_r_ (mm)
IA0	685.2	0.904	424.8	0.243	363.9	2.893
IA1	678.8	0.967	481.8	0.348	413.8	2.321
IA2	572.8	1.013	286.4	0.196	471.3	1.925
IA3	464.8	1.169	125.5	0.080	338.3	2.221
IA4	375.6	1.220	90.2	0.153	289.9	1.952
HIA0	559	1.028	212.4	0.137	325.1	2.980
HIA1	535.7	1.055	225.0	0.175	331.6	2.321
HIA2	455.7	1.215	259.7	0.379	320.4	2.552
HIA3	474.5	1.135	251.5	0.346	373.3	1.930
HIA4	406.1	1.370	101.5	0.170	338.3	2.466
NA0	159.3	0.946	113.1	0.380	95.9	2.458
NA1	312.4	0.842	203.0	0.216	190.8	2.007
NA2	1437.3	0.984	1078.2	0.378	714.6	2.792
NA3	2755.2	1.020	2176.0	0.503	1569.6	2.672
NA4	3550.0	0.674	3301.5	0.607	-	-

**Table 4 materials-15-06024-t004:** The nominal strength ratios.

Series	IA0	IA1	IA2	IA3	IA4
*u*_sf_ (%)	75.9	75.2	63.4	51.5	41.6
*u*_de_ (%)	58.5	67.2	46.5	24.5	21.9
*u*_res_ (%)	64.7	69.5	89.7	80.6	82.8
**Series**	**HIA0**	**HIA1**	**HIA2**	**HIA3**	**HIA4**
*u*_sf_ (%)	61.9	59.3	50.5	52.5	45.0
*u*_de_ (%)	35.1	38.9	52.9	49.4	22.5
*u*_res_ (%)	70.7	70.0	80.3	84.9	92.9
**Series**	**NA0**	**NA1**	**NA2**	**NA3**	**NA4**
*u*_sf_ (%)	70.6	69.2	70.8	76.3	-
*u*_de_ (%)	67.5	61.5	71.1	75.4	-
*u*_res_ (%)	69.7	68.2	59.5	67.1	-

**Table 5 materials-15-06024-t005:** Bond works.

Series	IA0	IA1	IA2	IA3	IA4
*W*_d_ (N∙mm)	59	78	32	5	8
*W*_p_ (N∙mm)	465	469	438	407	308
*W*_r_ (N∙mm)	1543	1233	892	795	556
**Series**	**HIA0**	**HIA1**	**HIA2**	**HIA3**	**HIA4**
*W*_d_ (N∙mm)	12	20	54	40	89
*W*_p_ (N∙mm)	422	408	387	349	386
*W*_r_ (N∙mm)	1276	967	893	691	794
**Series**	**NA0**	**NA1**	**NA2**	**NA3**	**NA4**
*W*_d_/*n* (N∙mm)	25	11	28	40	37
*W*_p_/*n* (N∙mm)	108	101	117	123	--
*W*_r_/*n* (N∙mm)	292	239	337	352	--

## Data Availability

Data are available with the first author and can be shared with anyone upon reasonable request.

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
