# Peer review of "Bond Behaviors of Steel Fiber in Mortar Affected by Inclination Angle and Fiber Spacing"

_materials, 2022, doi:10.3390/ma15176024_

Round 1
Reviewer 1 Report
Journal
Materials
Manuscript ID
materials-1838406
Title
Bond behaviors of steel fiber in mortar affected by inclination angle and fiber spacing
Comments
In this article, the authors studied three series of pullout tests for the hook-end steel fibres embedded in manufactured sand mortar to verify the pullout behaviour of steel fibres in groups with the implications of the inclination angle and the fibre spacing. Two series are tested to evaluate the effect of steel fibres inclination angles, and one is tested to evaluate the effect of the number of aligned fibres. In addition, pullout parameters were analyzed.
The manuscript does not present new methods or innovative findings; however the three series of pullout tests are of some interest, and the study can contribute to consolidating existing knowledge.
Add to the results:
· Please add a figure with the pullout–slip curves of specimens of Series (IA0, HIA0 and NA0), for each of four specimens in the same group, and with the averaged plot (add on the plot all the key points); in order to compare between series, the Pullout load must be divided by the fibre number.
· The maximum fibre stress σmax is an important parameter to estimate the fibre utilization efficiency and predicts fibre pullout modes, i.e. pullout or rupture. It also gives reference to determining the minimum fibre strength needed to avoid fibre fracture. Please add a figure or table and add a comment on the results.
· Evaluate the effect of mortar strengths on the results of specimens with the same inclination angle (IA0 and HIA0) and plot the results in figure 6c, and comment on it.
· Improve the discussion of the results in (3. Test Results and Analyses) by comparing them with other authors e.g. effect of inclination angels [21, 22] fibre group effect [27 and 28]
· Add these references in the introduction and comment/compare their results with yours:
https://doi.org/10.1016/j.conbuildmat.2022.127373
https://doi.org/10.1016/S0266-3538(02)00045-3
https://doi.org/10.1186/s40069-019-0344-1
https://doi.org/10.1617/s11527-009-9553-4
https://doi.org/10.1061/(ASCE)EM.1943-7889.0000800
Additional comments:
Add in line 111 the total number of specimens (dog-bone shape) used in the study.
Add in line 145 the standard deviation values
Figure 5(a, b and c) plotting most be in different colours with different line styles and/or markers and change the x-axis (slipe s) limits to 4 mm.
Figure 5(a, b) change the y-axis (Pullout load) limits max to 700 N.
Figure 6(a, b and c) change the plot ylabel to “strength (MPa)” and change the y-axis limits max to 10 MPa.
Figure 6(a, b and c) remove the second y-axes (strength ratio) and use a table to present the results.
Table 4 use for all the results (all series) W/n instead of W.
Eq. (2) use sp instead of Sp, and introduce all symbols used in the equations.
Author Response
Dear reviewer
Thanks very much for your attention and the comments on paper materials-1838406. Those comments are all valuable and very helpful for revising and improving our paper, as well as the important guiding significance to our researches. Here, we have made extensive modification on the original manuscript, and carefully proof-read the manuscript to minimize typographical, grammatical, and bibliographical errors. The main revised parts are represented as Red. We also attached revised manuscript in the format of MS word for your approval. Here below is our description on revision according to the reviewers’ comments.
Part A (reviewer 1)
Comment 1: In this article, the authors studied three series of pullout tests for the hook-end steel fibres embedded in manufactured sand mortar to verify the pullout behaviour of steel fibres in groups with the implications of the inclination angle and the fibre spacing. Two series are tested to evaluate the effect of steel fibres inclination angles, and one is tested to evaluate the effect of the number of aligned fibres. In addition, pullout parameters were analyzed.
The manuscript does not present new methods or innovative findings; however the three series of pullout tests are of some interest, and the study can contribute to consolidating existing knowledge.
Response: Thank you for your confirmation.
Comment 2: Please add a figure with the pullout–slip curves of specimens of Series (IA0, HIA0 and NA0), for each of four specimens in the same group, and with the averaged plot (add on the plot all the key points); in order to compare between series, the Pullout load must be divided by the fibre number.
Response: Thanks for your kind reminder. Figure 2 with the pullout load-slip curves of specimens of Series (IA0 and HIA0), for each of four specimens in the same group, and with the averaged plot is presented in the revised manuscript. The bond strengths between series are also mainly talked. The values of τd, τmax and τres of IA0 and HIA0 are also presented in the Figure 6(c). Ignoring the effect of embedded length, comparing the values of τd, τmax and τres of IA0, HIA0, NA0, NA1 and NA2, all the bond strengths increase with the flexural strength of mortar. The result is consistent with the previous study [14]. it is unreasonable to use the single fiber pull-out load curve for HIA series, thus, the curves of four fibers are presented for IA and HIA series to comparison. The total fiber and single fiber pull-out load-slip curves are both drawn in the NA series.
Comment 3: The maximum fibre stress σmax is an important parameter to estimate the fibre utilization efficiency and predicts fibre pullout modes, i.e. pullout or rupture. It also gives reference to determining the minimum fibre strength needed to avoid fibre fracture. Please add a figure or table and add a comment on the results.
Response: Thanks for your kind reminder. The nominal fiber utilization efficiency usf is the maximum fiber stress σmax divide the fiber tensile strength.The values of which are added in the Table 4 and discussed in the revised manuscript.
Comment 4: Evaluate the effect of mortar strengths on the results of specimens with the same inclination angle (IA0 and HIA0) and plot the results in figure 6c, and comment on it.
Response: Thanks for your kind reminder. It has been revised. The values of τd, τmax and τres of IA0 and HIA0 are also presented in the Figure 7(c). Ignoring the effect of embedded length, comparing the values of τd, τmax and τres of IA0, HIA0, NA0, NA1 and NA2, all the bond strengths increase with the flexural strength of mortar. This regularity is consistent with the previous study [14].
Comment 5: Improve the discussion of the results in (3. Test Results and Analyses) by comparing them with other authors e.g. effect of inclination angels [21, 22] fibre group effect [27 and 28]
Response: Thanks for your kind reminder. It has been revised. the Pp decreased about 45.2%, while the peak-slip sp increased about 26.2%. The regularity is also reported by the reference [25]. the reduction rate of the bond strengths of multiply fibers compared with single fiber is smaller than that with reported in the references [33,34]. This may attribute to the different cementitious matrix and pull-out test method.
Comment 6: Add these references in the introduction and comment/compare their results with yours:
https://doi.org/10.1016/j.conbuildmat.2022.127373
https://doi.org/10.1016/S0266-3538(02)00045-3
https://doi.org/10.1186/s40069-019-0344-1
https://doi.org/10.1617/s11527-009-9553-4
https://doi.org/10.1061/(ASCE)EM.1943-7889.0000800
Response: Thanks, it has been added.
Comment 7: Add in line 111 the total number of specimens (dog-bone shape) used in the study.
Response: Thanks, it has been added.
Comment 8: Add in line 145 the standard deviation values
Response: Thanks, it has been added.
Comment 9: Figure 5(a, b and c) plotting most be in different colours with different line styles and/or markers and change the x-axis (slipe s) limits to 4 mm.
Response: Thanks, it has been revised.
Comment 10: Figure 5(a, b) change the y-axis (Pullout load) limits max to 700 N.
Response: Thanks, it has been revised.
Comment 11: Figure 6(a, b and c) change the plot ylabel to “strength (MPa)” and change the y-axis limits max to 10 MPa.
Response: Thanks, it has been revised.
Comment 12: Figure 6(a, b and c) remove the second y-axes (strength ratio) and use a table to present the results.
Response: Thanks, it has been revised.
Comment 13: Table 4 use for all the results (all series) W/n instead of W.
Response: Thanks for your kind reminder. It is unreasonable to use the W/n to evaluate the HIA series. Thus, only the NA series use W/n.
Comment 14: Eq. (2) use sp instead of Sp, and introduce all symbols used in the equations.
Response: Thanks, it has been revised.
Reviewer 2 Report
The article presents the results of an extensive experimental program that includes several variables in the bond behavior of steel fibers in mortar. Some suggestions and questions are made to improve the manuscript:
1) In the introduction, the authors present studies that evaluated the bond behavior between steel fibers and concrete matrix, but in their experimental study, they evaluated it in mortars. How do the authors imagine that the presence of coarse aggregates could impact the results obtained?
2) Still on the previous question, the authors should include results from further studies on mortars in the introduction.
3) In the methodology, the authors report that the properties of the raw materials are present in another paper, but they do not explain the choice of using fly ash. Why did the authors use this material? How was the mix composition of these mortars made?
4) The reason for the difference in the mechanical behavior of the three series studied is not very clear in the text. Do they all have the same composition and test age? Why the difference observed?
5) The results and analysis need to be more in-depth. What are the results obtained in previous studies? Such a comparison is valid.
6) Although they are from different series, I think it's important to put the graphs in figure 5 on the same scale, at least (a) and (b).
7) The results presented in table 3 could be better discussed.
8) I recommend using the same scale for the charts in Figure 6.
9) The quality of figure 7 is too low. Authors should present it in a higher resolution or remove it from the article.
Author Response
Dear reviewer
Thanks very much for your attention and comments on paper materials-1838406. Those comments are all valuable and very helpful for revising and improving our paper, as well as the important guiding significance to our researches. Here, we have made extensive modification on the original manuscript, and carefully proof-read the manuscript to minimize typographical, grammatical, and bibliographical errors. The main revised parts are represented as Red. We also attached revised manuscript in the format of MS word for your approval. Here below is our description on revision according to the reviewers’ comments.
Part A (reviewer 2)
Comment 1: In the introduction, the authors present studies that evaluated the bond behavior between steel fibers and concrete matrix, but in their experimental study, they evaluated it in mortars. How do the authors imagine that the presence of coarse aggregates could impact the results obtained?
Response: Thanks for your kind reminder. Due to the geometric size of coarse aggregate and the steel fiber commonly used size are basically in the same order of magnitude, the influence of coarse aggregate on the steel fiber reinforced effect in SFRC is mainly reflected in its influence on the fiber distribution. Thus, in my opinion, it is acceptable to evaluate the bond performance between fiber and concrete using the pull-out test of fiber and mortar paste in concrete.
Comment 2: Still on the previous question, the authors should include results from further studies on mortars in the introduction.
Response: Thanks for your kind reminder. It needs to be clarified that part of the literature on bonding properties discussed in the abstract is based on mortar. The author rechecked the abstract and further explained it in the revised manuscript.
Comment 3: In the methodology, the authors report that the properties of the raw materials are present in another paper, but they do not explain the choice of using fly ash. Why did the authors use this material? How was the mix composition of these mortars made?
Response: Thanks for your kind reminder. In order to ensure the accuracy of the fibers position in the mortar specimen, the self-compacting workability is needed for the fresh mortar. Thus, fly ash is added as mineral admixture and water reducer is used. A planetary type mortar mixer was used for the mortar. Water with water reducer, the cement and fly ash were firstly added in the mixing pot, then start mixing for 30 sec. Then, adding manufactured sand evenly at the beginning of the second 30 sec. At last, mixing for another 30 sec.
Comment 4: The reason for the difference in the mechanical behavior of the three series studied is not very clear in the text. Do they all have the same composition and test age? Why the difference observed?
Response: Yes, the three series studied have the same composition and test age. However, due to the limited number of test moulds, the three series specimens are not poured on the same day. The ambient temperature and humidity of specimens at the first 2 days are different. Besides, the intrinsic discreteness is existing for the mechanical properties of mortar. Thus, the compressive and the subsequent bonding performance strength of the three series studies, especially the flexural strength, are different.
Comment 5: The results and analysis need to be more in-depth. What are the results obtained in previous studies? Such a comparison is valid.
Response: Thanks for your kind reminder. It has been revised. The results obtained in previous studies has been presented in line 93-104 in the introduction of the revised manuscript. This study is a follow-up study based on the above studies, so there is no comparison.
Comment 6: Although they are from different series, I think it's important to put the graphs in figure 5 on the same scale, at least (a) and (b).
Response: Thanks, it has been revised.
Comment 7: The results presented in table 3 could be better discussed.
Response: Thanks for your kind reminder. It has been revised. With the inclination angle increased from 15° to 60°, the slope of the ascending portion decreases, the Pp decreased about 45.2%, while the peak-slip sp increased about 26.2%. The regularity is also reported by the reference [25]. The Pd increased 13.4% with the inclination angel increase to 15°, then obviously decreased 78.8% with the angel continued increase to 60°.The Pr increased 29.5% with the inclination angel increase to 30°, then decreased 20.3% with the angel continued increase to 60°. Both slips sd and sr show decrease regularity with the increased inclination angle with a certain variation.
Comment 8: I recommend using the same scale for the charts in Figure 6.
Response: Thanks, it has been revised.
Comment 9: The quality of figure 7 is too low. Authors should present it in a higher resolution or remove it from the article.
Response: Thanks, it has been revised.
Reviewer 3 Report
The manuscript analyzes the effect of steel fiber distribution on the characteristics of the behavior of mortars reinforced with steel fibers. Although the article is quite remarkable with its aim and structure, it contains a lot of mistakes in terms of language usage. It is recommended that the article be reviewed by a native speaker in order to be considered for publication.
Author Response
MS Type: Article
Manuscript number: materials-1838406
Title: Bond behaviors of steel fiber in mortar affected by inclination angle and fiber spacing
Correspondence Author: Xinxin Ding, Shunbo Zhao
Comment: The manuscript analyzes the effect of steel fiber distribution on the characteristics of the behavior of mortars reinforced with steel fibers. Although the article is quite remarkable with its aim and structure, it contains a lot of mistakes in terms of language usage. It is recommended that the article be reviewed by a native speaker in order to be considered for publication.
Response: Thanks very much for your comments on paper materials-1838406. Here, we have made extensive modification on the original manuscript, and carefully proof-read the manuscript to minimize typographical, grammatical, and bibliographical errors. The main revised parts are represented as Red.
Reviewer 4 Report
Kindly avoid the first person singular and plural like “I”, “we”, “us” etc. Thanks
What is the reason for adapting dog-bone-shaped specimens?
Line 120: Kindly review the Lsf? Why the spacing is not presented in some ascending or descending order? Secondly, 0, 3.5, 5, 7, and then 22.1 mm of spacing are chosen. Any specific reason.
Section 2.3: Why the specimens were placed in molds for 48 hours instead of 24.
Refer to Figure 2: It is mathematically obvious that spalling of mortar may take place due to an increase in inclination angle. Figure 2: The peeling off mortar seems more pronounced at a 45º angle than at a 60º angle.
Figure 3: The individual behavior of the straight and inclined fibers may be highlighted in the Figure.
Figure 4: The effect of spacing is not clear. What is the effect of spacing on the response of the specimen? Is it random?
Figure 5 shows brittle behavior for NA4. Reason?
Last sentence of conclusions: “Besides, the number of steel fiber in mortar should be no less than 4 to ensure the validity of pullout test for the bond performance of steel fiber.” I could not find this conclusion in the text. Can you please refer to where you got this conclusion?
The effect of number of fibers was examined through 1, 2, 9, 16, and 25 fibers. There is no specific order. “1” or “2” are too less and “25” is too much in the given space. And then there is no “4” in the number, which was declared as the threshold value. Kindly enlighten me about that. 4, 9, and 16 were the more appropriate choices.
Author Response
Dear reviewer
Thanks very much for your attention and comments on paper materials-1838406. Those comments are all valuable and very helpful for revising and improving our paper, as well as the important guiding significance to our researches. Here, we have made extensive modification on the original manuscript, and carefully proof-read the manuscript to minimize typographical, grammatical, and bibliographical errors. The main revised parts are represented as Red. We also attached revised manuscript in the format of MS word for your approval. Here below is our description on revision according to the reviewers’ comments.
Part A (reviewer 3)
Comment 1: Kindly avoid the first person singular and plural like “I”, “we”, “us” etc. Thanks
Response: Thanks for your kind reminder. It has been revised.
Comment 2: What is the reason for adapting dog-bone-shaped specimens?
Response: Thanks. dog-bone-shaped specimens is adapting refer to the pullout specimen design in the specification of China standard CECS 13.
Comment 3: Line 120: Kindly review the Lsf? Why the spacing is not presented in some ascending or descending order? Secondly, 0, 3.5, 5, 7, and then 22.1 mm of spacing are chosen. Any specific reason.
Response: Thanks. It has been clarified further in the Figure 1(c) and line 143-144 of the revised manuscript. Figure 1(c) presents the geometric details of specimen for Series NA with different number of aligned steel fibers. The area of 16mm × 16 mm in the center of the cross section is designed as the area for fiber placement. The fiber center-to-center spacing Lsf is corresponds to the designed fiber number.
Comment 4: Section 2.3: Why the specimens were placed in molds for 48 hours instead of 24.
Response: Thanks for your kind reminder. In order to minimize the disturbance of demolding on the bond between the fibers and the mortar matrix the specimen was demolded after cast for 2 days, and placed into the standard curing box for another 26 days before testing.
Comment 5: Refer to Figure 2: It is mathematically obvious that spalling of mortar may take place due to an increase in inclination angle. Figure 2: The peeling off mortar seems more pronounced at a 45º angle than at a 60º angle.
Response: Thanks for your kind reminder. It has been explained in line 198-201 in the revised manuscript. It should be declared that t Although a similar peeling off area of the mortar presented on specimens with steel fibers inclined at angle 45° and 60°, a larger peeling off depth happened on the specimens with steel fibers at greater inclination angle.
Comment 6: Figure 3: The individual behavior of the straight and inclined fibers may be highlighted in the Figure.
Response: Thanks for your kind reminder. The spalling areas of mortar near the straight and inclined fibers are highlighted as red and blue lines, respectively.
Comment 7: Figure 4: The effect of spacing is not clear. What is the effect of spacing on the response of the specimen? Is it random?
Response: No, it is not random. The effect of spacing is obviously on the pull-out load-slip curves and the indexes τd, τmax, τres, ude and ures.When the fiber number reached to 16 for NA3, the τd, τmax and τres increase by 20.2%, 8.1% and 2.3% compared with those of NA0, respectively. This indicates that a group effect of parallel fibers with fiber spacing no less than 5 mm benefits to the bond performances. Therefore, an interaction exists among steel fibers in concrete matrix if the steel fibers are uniformly distributed in parallel with a volume fraction over 0.78% (corresponding to Lsf of 5mm).
Comment 8: Figure 5 shows brittle behavior for NA4. Reason?
Response: The reason of brittle behavior for NA4 is talked in line 220-222 in the revised manuscript. the mortars of NA4 peeled off accompanied with slightly straightened hook-end of steel fibers. The tensile strength of the mortar near the fiber end is not enough to resist the stress transmitted by the fiber to the matrix. This indicates a rational spacing among steel fibers is necessary to ensure a sufficient surrounding mortar which can provides anchorage shear resistance of the interface between steel fiber and mortar.
Comment 9: Last sentence of conclusions: “Besides, the number of steel fiber in mortar should be no less than 4 to ensure the validity of pullout test for the bond performance of steel fiber.” I could not find this conclusion in the text. Can you please refer to where you got this conclusion?
Response: Thanks for your kind reminder. It has been revised as “specimen with fiber number from 4 to 9 in this study are suitable for study the bond mechanism between the fiber and mortar matrix. Comprehensively considering the range and test accuracy of the test system in this study, four steel fibers symmetrically arranged to section centroid is a better of chose.”
Comment 10: The effect of number of fibers was examined through 1, 2, 9, 16, and 25 fibers. There is no specific order. “1” or “2” are too less and “25” is too much in the given space. And then there is no “4” in the number, which was declared as the threshold value. Kindly enlighten me about that. 4, 9, and 16 were the more appropriate choices.
Response: Thanks for your kind reminder. The NA series tests were designed to verify the pullout behavior of steel fibers in groups. Thus, the minimum number of fiber and as much number of fibers as possible is designed in this study. NA0 with one fiber is the referenced one. The pull-out load-slip curve and τd, τmax, τres, ude and ures of NA1 with 2 fibers, NA2 with 9 fibers show no significant difference. When the fiber number reached to 16 for NA3, the τd, τmax and τres increase by 20.2%, 8.1% and 2.3% compared with those of NA0, respectively. This indicates that a group effect of parallel fibers with fiber spacing no less than 5 mm benefits to the bond performances. On the other point of view, specimen with fiber number from 1 to 9 are suitable for study the bond mechanism between the fiber and mortar matrix. Comprehensively considering the range and test accuracy of the test system in this study, four steel fibers symmetrically arranged to section centroid is a better of chose.
Round 2
Reviewer 1 Report
The authors have made all the required revisions, and this manuscript could be accepted for publication in its revised form.
Reviewer 2 Report
Authors have made all the required revisions and this manuscript could be accepted for now.
Reviewer 3 Report
The manuscript is acceptable for publication in its revised form.